# Synthesis and Anti-*Trypanosoma cruzi* Biological Evaluation of Novel 2-Nitropyrrole Derivatives

**DOI:** 10.3390/molecules27072163

**Published:** 2022-03-27

**Authors:** Fanny Mathias, Youssef Kabri, Damien Brun, Nicolas Primas, Carole Di Giorgio, Patrice Vanelle

**Affiliations:** 1Equipe Pharmaco-Chimie Radicalaire, CNRS, ICR UMR 7273, Faculté de Pharmacie, Aix Marseille University, 27 Boulevard Jean Moulin, CS30064, CEDEX 05, 13385 Marseille, France; fanny.mathias@univ-amu.fr (F.M.); youssef.kabri@univ-amu.fr (Y.K.); brundamien83@gmail.com (D.B.); nicolas.primas@univ-amu.fr (N.P.); 2Assistance Publique-Hôpitaux de Marseille (APHM), Pharmacie Usage Intérieur, Hôpital Nord, Chemin-des-Bourrely, 13015 Marseille, France; 3Assistance Publique-Hôpitaux de Marseille (APHM), Service Central de la Qualité et de l’Information Pharmaceutiques (SCQIP), Hôpital de la Conception, 147, Boulevard Baille, 13005 Marseille, France; 4Laboratoire de Mutagénèse Environnementale, CNRS, IRD, Aix Marseille University, IMBE UMR 7263, Avignon University, 13385 Marseille, France; carole.di-giorgio@univ-amu.fr

**Keywords:** chagas disease, *T. cruzi*, pyrrole, nitro-heterocycle

## Abstract

Human American trypanosomiasis, called Chagas disease, caused by *T. cruzi* protozoan infection, represents a major public health problem, with about 7000 annual deaths in Latin America. As part of the search for new and safe anti-*Trypanosoma cruzi* derivatives involving nitroheterocycles, we report herein the synthesis of ten 1-substituted 2-nitropyrrole compounds and their biological evaluation. After an optimization phase, a convergent synthesis methodology was used to obtain these new final compounds in two steps from the 2-nitropyrrole starting product. All the designed derivatives follow Lipinski’s rule of five. The cytotoxicity evaluation on CHO cells showed no significant cytotoxicity, except for compound **3** (CC_50_ = 24.3 µM). Compound **18** appeared to show activity against *T. cruzi* intracellular amastigotes form (EC_50_ = 3.6 ± 1.8 µM) and good selectivity over the vero host cells. Unfortunately, this compound **18** showed an insufficient maximum effect compared to the reference drug (nifurtimox). Whether longer duration treatments may eliminate all parasites remains to be explored.

## 1. Introduction

*Trypanosoma cruzi* is a flagellate protozoan involved in Chagas disease (Human American trypanosomiasis), a neglected tropical disease causing a major public health problem, with about 7000 annual deaths in Latin America [1]. Although asymptomatic in the acute phase after contamination, cardiac and digestive complications may appear in 20 to 30% of cases during the chronic phase of the disease (inflammatory cardiomyopathy, enlargement of the gastrointestinal tract and organs, and gastrointestinal motor disorders) [2,3].

The anti-trypanosomatid potential of nitroheterocycles is well known, such as benznidazole (2-nitroimidazole) and nifurtimox (5-nitrofuran) listed on the WHO Model list of essential medicines for the treatment of human American trypanosomiasis (Figure 1) [4]. These two compounds represent an important part of the therapeutic arsenal in the management of Chagas disease. However, they are active mainly in the acute and early chronic stage of the disease, but not once in the late stage [5]. Their use is also limited by their toxicity, making the development of new safe and effective drugs against *T. cruzi* necessary. Recently, an orally available 5-nitroimidazole, fexinidazole, was identified as a promising new drug candidate for the treatment of both stages of human African trypanosomiasis due to *Trypanosoma brucei gambiense* infection [6]. The clinical evaluation of this compound on the treatment of adults with chronic indeterminate Chagas disease due to *T. cruzi* infection is in progress [7,8].

On the other hand, several research studies were carried out on heterocyclic nitrogen mono-cyclic (imidazole, thiazole, triazole) or bicyclic (benzimidazole, imidazopyridine, imidazooxazole, imidazooxazine) series with several mechanisms of action identified against *T. cruzi* [9,10,11,12]. Only a few recent publications have shown that pyrrole derivatives could have good anti-*Trypanosoma cruzi* activity by inhibition of an enzyme involved in the synthesis of sterols, 14α-demethylase, also known as CYP51 [13].

The aim of this work was to synthesize 2-nitropyrrole derivatives functionalized at the N1-position by a chain structurally close to that found in benznidazole (Figure 2). Several nitro-drugs are known to act as prodrugs bioactivated by *T. cruzi* nitro-reductases NTR type I [14,15], capable of exerting cytotoxic effects via DNA damage. We will investigate wether the synthesized 2-nitropyrrole derivatives exhibit anti-*T. cruzi* activity.

## 2. Results and Discussion

### 2.1. Chemistry

The regioselective nitration of the commercial pyrrole using previous report allows us to synthesize 2-nitropyrrole starting product in 60% yield (Figure 1) [16].

The X-ray crystal structure (Figure 3) was determined for this compound to confirm the regioselective nitration reaction at the 2-position of the pyrrole compound (CCDC 2132900).

The pyrrole nitration must be carried out under mild conditions using acetic anhydride as a solvent, because in the presence of sulfuric acid, only the polymerization of the pyrrole was reported, without the formation of the target product [17].

In the reaction conditions described in Figure 1, we observed the formation of the expected product (**1**) in good yield, traces of the dinitro pyrrole product observed by LC/MS monitoring but not isolated, as well as a black resin which may correspond to a polymerization side-product, as reported in the literature [17].

Then, we tried two alkylation protocols to obtain the 2-nitropyrrole derivatives functionalized at position N-1.

First, we applied the benznidazole alkylation protocol on the 2-nitropyrrole starting product in two steps (Figure 2) [18].

The first alkylation step with ethyl bromoacetate led to the desired compound **2** in a 71% yield (Figure 2, step 1). Unfortunately, the second amidification step with benzylamine and aniline did not lead to the desired compounds **3** or **4** (Figure 2, step 2). LC/MS monitoring of this reaction revealed the disappearance of the starting product signal and the absence of formation of the desired products **3** or **4**. A reaction optimization using LC/MS monitoring was therefore conducted by varying different parameters according to amidification protocols described in the literature (Table 1) [19]. The various attempts were not conclusive, and we never observed the expected product **3** or **4** by LC/MS monitoring. The formation of a black resin was visually observed for attempts 1, 4 and 5. It was described in the literature that the reactions on pyrrole consuming the starting product, without the formation of the desired product or other side products, suggested a pyrrole polymerization because of its high reactivity [17].

In the first step, we showed that the pyrrole alkylation with ethyl bromoacetate was successfully conducted (Figure 2, step 1). The limiting step was the amidification reaction. We have therefore changed the procedure and decided to use a convergent synthesis methodology: performing, on one hand, the acylation of various primary amine to form bromoacetamide intermediates and carry out the substitution reaction on the pyrrole **1** in the last step. The acylation of commercially available 2-bromoacetyl bromide was successfully carried out with 10 different amines using the experimental protocol reported in the literature (Table 2) [20].

Secondly, we performed a nucleophilic substitution reaction between compound **1** and bromoacetamide intermediates (**5**–**14**) to obtain the 2-nitropyrrole derivatives functionalized in 1-position (Table 3). This synthetic methodology was applied to the various bromoacetamide intermediates **5** to **14** to obtain final compounds **3–4** and **15**–**22** in good yields ranging from 58% to 96%, except for compound **20** obtained in a lower yield of 30%. Under these conditions, we did not observe any problem related to the polymerization of pyrrole.

### 2.2. Biological Evaluation

#### 2.2.1. Cytotoxicity and anti-*T. cruzi* Biological Evaluation

These new derivatives were evaluated in vitro against the intracellular *T. cruzi* strain X10/7-Silvio by determining their 50% efficacy concentration (EC_50_) and were compared to the reference anti-*T. cruzi* drug nifurtimox. The *in vitro* 50% cytotoxic concentrations (CC_50_) were assessed on the CHO cell line and compared to a cytotoxic reference drug, doxorubicin. *In vitro* cytotoxicity against vero host cells was also assessed (vero cells EC_50_). The results are presented in Table 4.

Regarding the effect on cell viability of the CHO cells, except for one compound (**3**), the other derivatives showed good cytotoxicity values ranging from 155.6 to 321.1 µM compared to the reference drug, doxorubicin (CC_50_ CHO = 4.6 µM). Three compounds showed no cytotoxicity against vero host cells at concentrations up to 50 µM (**17**–**19**).

Of the 10 synthesized molecules, three showed poor DMSO solubility, limiting their *in vitro* anti-*T. cruzi* evaluation (**20**–**22**). Three compounds showed *T. cruzi* EC_50_ < 10 µM (**4**, **16** and **18**). Only compound **18** exhibited activity against the parasite (EC_50_ = 3.6 ± 1.8 µM) without toxicity against host cells at concentrations up to 50 µM. However, the maximum effect of the latter compound (Figure 4) compared to the nifurtimox control (Figure 5) was relatively low (79% and 71% in two independent replicates), indicating that a substantial number of parasites survive 96h treatment. Whether longer duration treatments may eliminate all parasites remains to be explored.

The structure–activity relationship (SAR) can be suggested from these biological results. The presence of a benzyl group seems to be unfavourable for anti-*T. cruzi* activity. Indeed, compounds **3** and **15** with respectively benzyl and *meta*-EWG 3-CF_3_ benzyl group exhibit no activity against *T. cruzi* (EC_50_ = 34.5 µM and 41.0 µM), whereas their analogue with phenyl (**4**) and meta-EWG 3-CF_3_ phenyl group (**18**) exhibits activity (EC_50_ < 10 µM). The meta-EWG 3-CF_3_ phenyl group gave the best anti-*T. cruzi* activity (**18**), while the *meta*-electro-donating OCH_3_ phenyl group (**19**) gave the lowest anti-*T. cruzi* activity (EC_50_ = 50.0 µM). An electron-withdrawing (EWG) group in the *meta*-position of the phenyl seems to be important for anti-*T. cruzi* activity.

#### 2.2.2. In Silico ADME Properties

The ADME properties of 2-nitropyrrole derivatives (**3**–**4**, **15**–**22**) were predicted by using molinspiration [21]. The designed derivatives follow Lipinski’s rule of five (Table 5) [22]. There are less than 5 H-bond donors, the molecular weight is less than 500 g·mol^−1^, the logP is less than 5, and there are less than 10 H-bond acceptors. In addition, the TPSA was <140 Å^2^ (Veber’s rules). There are no parameters out of range, so we did not identify any alerts of poor absorption or permeability for the oral route of administration.

Compliance with Lipinski’s rule of five is reassuring, but we must keep in mind that not all drugs fully comply with this rule, especially in the antimicrobial field [23].

## 3. Materials and Methods

### 3.1. Chemistry

#### 3.1.1. Generality

Melting points were determined on a Köfler melting point apparatus (Wagner & Munz GmbH, München, Germany) and were uncorrected. NMR spectra were recorded on AV 250 spectrometers or a Bruker Avance NEO 400 MHz NanoBay spectrometer at the Faculté de Pharmacie of Marseille or on a Bruker Avance III nanobay 400 MHz spectrometer at the Spectropole, Faculté des Sciences de Saint-Jêrome (Marseille). (1H NMR: reference CHCl_3_ δ = 7.26 ppm, reference DMSO-d_6_ δ = 2.50 ppm and 13C NMR: reference CHCl_3_ δ = 76.9 ppm, reference DMSO-d_6_ δ = 39.52 ppm). The following adsorbent was used for column chromatography: silica gel 60 (Merck KGaA, Darmstadt, Germany, particle size 0.063–0.200 mm, 70–230 mesh ASTM). TLC was performed on 5 cm × 10 cm aluminium plates coated with silica gel 60F-254 (Merck) in an appropriate eluent. Visualisation was performed with ultraviolet light (234 nm). The purity of synthesised compounds was checked by LC/MS analyses, which were realized at the Faculté de Pharmacie of Marseille with a Thermo Scientific Accela High Speed LC System^®®^ (Waltham, MA, USA) coupled using a single quadrupole mass spectrometer Thermo MSQ Plus^®®^. The RP-HPLC column is a Thermo Hypersil Gold^®®^ 50 × 2.1 mm (C18 bounded), with particles of a diameter of 1.9 mm. The volume of sample injected on the column was 1 µL. Chromatographic analysis, total duration of 8 min, was on the gradient of the following solvents: t = 0 min, methanol/water 50:50; 0 < t < 4 min, linear increase in the proportion of methanol to a methanol/water ratio of 95:5; 4 < t < 6 min, methanol/water 95:5; 6 < t < 7 min, linear decrease in the proportion of methanol to return to a methanol/water ratio of 50:50; 6 < t < 7 min, methanol/water 50:50. The water used was buffered with ammonium acetate 5 mM. The flow rate of the mobile phase was 0.3 mL/min. The retention times (tR) of the molecules analysed were indicated in min. High-resolution MS experiments were performed at the Spectropole (Campus Etoile, Aix-Marseille Université, France) with a SYNAPT G2 HDMS quadrupole/time-of-flight (Q/ToF) (Waters, Manchester, UK) using the following parameters: ESI capillary voltage: +2.8 kV; extraction cone voltage: 20 V; and desolvation gas flow rate (nitrogen): 100 L·h^−1^. The accurate mass measurements were carried out in triplicate with an external calibration. Reagents were purchased and used without further purifications from Sigma-Aldrich or Fluorochem. The starting product 2-nitropyrrole (1) was prepared using previous reports [16]. Please see Appendix A for details.

#### 3.1.2. Synthesis of Ethyl 2-(2-nitro-1H-pyrrol-1-yl)acetate (2)

To a solution of 2-nitropyrrole (0.5 g, 4.76 mmol) in EtOH (20 mL) was added K_2_CO_3_ (0.679 g, 4.91 mmol, 1.1 eq) and ethyl bromoacetate (0.544 mL, 4.91 mmol, 1.1 eq). The reaction mixture was heated at 70 °C for 12 h. After EtOH evaporation, 20 mL of water was poured on the crude product and the aqueous layer was extracted with dichloromethane (3 × 50 mL). The organic layer was washed with brine (3 × 100 mL), dried over Na_2_SO_4_, and evaporated. The crude product was purified by column chromatography (silica gel, petroleum ether/ethyl acetate 9:1).

Yield 71% (629 mg). White solid. Mp 51–52 °C. 1H NMR (400 MHz, CDCl_3_) δ 7.26–7.24 (m, 1H, CH), 6.81 (dd, 3*J*_H-H_ = 2.8 Hz, 4*J*_H-H_ = 2.1 Hz, 1H, CH), 6.25 (dd, 3*J*_H-H_ = 4.4 Hz, 4*J*_H-H_ = 2.8 Hz, 1H, CH), 5.02 (s, 2H, CH_2_), 4.24 (q, 3*J*_H-H_ = 7.1 Hz, 2H, CH_2_), 1.28 (t, 3*J*_H-H_ = 7.1 Hz, 3H, CH_3_). 13C NMR (100 MHz, CDCl_3_) δ 167.5 (C), 137.7 (C), 130.0 (CH), 114.9 (CH), 109.2 (CH), 62.2 (CH_2_), 51.7 (CH_2_), 14.2 (CH_3_).

#### 3.1.3. General Procedure for the Synthesis of Bromoacetamide Intermediate Derivatives

In a first vial, dichloromethane (10 mL) was added to bromoacetyl bromide (216 µL, 2.48 mmol, 1 eq). In a second vial, dichloromethane (10 mL) was added to the corresponding amine (2.48 mmol, 1 eq) and Et_3_N (345 µL, 2.48 mmol, 1 eq). The two reactions mixtures were respectively stirred for 5 min at 0 °C in a bath of ice. The second solution mixture was added dropwise at 0 °C to the first bromoacetyl bromide solution, and the reaction mixture was stirred from 0 °C to room temperature for 12 h. The solution was poured into a bath of ice. The aqueous layer was extracted with dichloromethane (3 × 100 mL) and the organic layer was washed with brine (3 × 100 mL), dried over Na_2_SO_4_, and evaporated. Intermediate compounds were obtained without purification, except for compounds 7, 10, 11, 14.

*N*-Benzyl-2-bromoacetamide (**5**) [24]

Yield 82% (460 mg). White solid. Lit. Mp 106–107.5 °C [25]. 1H NMR (400 MHz, DMSO-d6) δ 8.78 (s, 1H, NH), 7.35–7.31 (m, 2H, 2CH), 7.27–7.25 (m, 3H, 3CH), 4.30 (d, 3*J*_H-H_ = 6.0 Hz, 2H, CH_2_), 3.91 (s, 2H, CH_2_). The 1H NMR data were in agreement with the literature values. Lit. 13C NMR (125 MHz, CDCl_3_) δ 165.3 (C),137.2 (C), 128.8 (2CH), 127.8 (CH), 127.7 (2CH), 44.2 (CH_2_), 29.1 (CH_2_).

2-Bromo-*N*-phenylacetamide (**6**) [24,26]

Yield 99% (525 mg). Brown solid. Lit. Mp 129–131 °C (isopropyl ether) [26]. 1H NMR (400 MHz, DMSO-d6) δ 10.37 (s, 1H, NH), 7.60–7.57 (m, 2H, 2CH), 7.35–7.30 (m, 2H, 2CH), 7.10–7.07 (m, 1H, CH), 4.04 (s, 2H, CH_2_). The 1H NMR data were in agreement with the literature values. Lit. 13C NMR (125 MHz, CDCl_3_) δ 163.3 (C), 136.9 (C), 129.1 (2CH), 125.2 (2CH), 120.0 (CH), 29.5 (CH_2_).

2-Bromo-*N*-[3-(trifluoromethyl)benzyl]acetamide (**7**) [27]

The crude product was purified by column chromatography (silica gel, dichloromethane/petroleum ether from 2/8 to 8/2). Yield 18% (130 mg). Brown solid. Lit. Mp 251–253 °C [28]. 1H NMR (400 MHz, CDCl_3_) δ 7.57–7.53 (m, 2H, 2CH), 7.49–7.47 (m, 2H, 2CH), 6.86 (s, 1H, NH), 4.54 (d, 3*J*_H-H_ = 6.1 Hz, 2H, CH_2_), 3.95 (s, 2H, CH_2_). The 1H NMR data were in agreement with the literature values. Lit. 13C NMR (CDCl_3_, 75 MHz) δ 165.6 (C), 141.4 (C), 130.0 (q, 3*J*_C-F_ = 32.3 Hz, C), 127.8 (2CH), 125.7 (q, 4*J*_C-F_ = 3.8 Hz, CH), 122.1 (CH), 118.6 (C), 43.6 (CH_2_), 28.9 (CH_2_).

2-Bromo-*N*-(4-bromophenyl)acetamide (**8**) [29]

Yield 91% (657 mg). Brown solid. Lit. Mp 148–150 °C. 1H NMR (400 MHz, DMSO-d6) δ 10.51 (s, 1H, NH), 7.57–7.50 (m, 4H, 4CH), 4.03 (s, 2H, CH_2_). The 1H NMR data were in agreement with the literature values. Lit. 13C NMR (CDCl_3,_ 100 MHz,) δ 163.3 (C), 136.0 (C), 132.1 (2CH), 121.5 (2CH), 117.9 (C), 29.3 (CH_2_).

2-Bromo-*N*-(2-chlorophenyl)acetamide (**9**)

Yield 44% (657 mg). Brown solid. Mp 92–93 °C. 1H NMR (400 MHz, DMSO-d6) δ 9.94 (s, 1H, NH), 7.71 (dd, 3*J*_H-H_ = 6.1 Hz, 4*J*_H-H_ = 1.7 Hz, 1H, CH), 7.52 (dd, 3*J*_H-H_ = 8.0 Hz, 4*J*_H-H_ = 1.5 Hz, 1H, CH), 7.35 (td, 3*J*_H-H_ = 7.8 Hz, 4*J*_H-H_ = 1.5 Hz, 1H, CH), 7.23 (td, 3*J*_H-H_ = 7.7 Hz, 4*J*_H-H_ = 1.7 Hz, 1H, CH), 4.17 (s, 2H, CH_2_). The 1H NMR data were in agreement with the literature values. 13C NMR (100 MHz, DMSO-d6) δ 170.7 (C), 134.2 (C), 129.3 (CH), 127.9 (2CH), 125.3 (CH), 122.0 (C), 61.6 (CH_2_).

2-Bromo-*N*-[3-(trifluoromethyl)phenyl]acetamide (**10**) [28]

The crude product was purified by column chromatography (silica gel, dichloromethane/petroleum ether 6/4). Yield 48% (331 mg). White solid. Lit. Mp 78–81 °C. 1H NMR (400 MHz, DMSO-d6) δ 10.72 (s, 1H, NH), 8.07 (s, 1H, CH), 7.76 (d, 3*J*_H-H_ = 8.4 Hz, 1H, CH), 7.60–7.56 (m, 1H, CH), 7.44 (d, 3*J*_H-H_ = 7.6 Hz, 1H, CH), 4.07 (s, 2H, CH_2_). The 1H NMR data were in agreement with the literature values. Lit. 13C NMR (100 MHz, CDCl_3_): δ 173.7 (C), 138.4 (C), 130.9 (C), 130.6 (CH), 127.8 (CH), 124.6 (C), 123.5 (CH), 123.1 (CH), 42.7 (CH_2_).

2-Bromo-*N*-(3-methoxyphenyl)acetamide (**11**) [30]

The crude product was purified by column chromatography (silica gel, dichloromethane/petroleum ether 6/4). Yield 46% (280 mg). White solid. Mp 94–95 °C. 1H NMR (400 MHz, CDCl_3_) δ 8.10 (s, 1H, NH), 7.27–7.26 (m, 1H, CH), 7.25–7.23 (m, 1H, CH), 7.02–6.99 (m, 1H, CH), 6.73–6.71 (m, 1H, CH), 4.02 (s, 2H, CH_2_), 3.81 (s, 3H, CH_3_). The 1H NMR data were in agreement with the literature values. Lit. 13C NMR (63 MHz, DMSO-d6) δ 164.7 (C), 159.5 (C), 139.7 (C), 129.6 (CH), 111.5 (CH), 109.2 (CH), 105.0 (CH), 55.0 (CH_3_), 30.4 (CH_2_).

2-Bromo-*N*-isopropylacetamide (**12**) [31]

Yield 25% (100 mg). White solid. Mp Lit. 63–64 °C. 1H NMR (400 MHz, CDCl_3_) δ 6.29 (s, 1H, NH), 4.10–4.02 (m, 1H, CH), 3.85 (s, 2H, CH_2_), 1.19 (d, 3*J*_H-H_ = 6.5 Hz, 6H, 2CH_3_). The 1H NMR data were in agreement with the literature values. Lit. 13C NMR (400 MHz, CDCl_3_) δ 164.4 (C), 42.3 (CH), 29.4 (CH_2_), 22.4 (2CH_3_).

2-Bromo-*N*-cyclopentylacetamide (**13**) [31]

Yield 87% (443 mg). White solid. Mp Lit. 84–85 °C. 1H NMR (400 MHz, CDCl_3_) δ = 6.74 (s, 1H, NH), 4.13–4.08 (m, 1H, CH), 3.78 (s, 2H, CH_2_), 1.94–1.87 (m, 2H, CH_2_), 1.65–1.63 (m, 2H, CH_2_), 1.56–1.53 (m, 2H, CH_2_), 1.40–1.36 (m, 2H, CH_2_). The 1H NMR data were in agreement with the literature values. Lit. 13C NMR (400 MHz, CDCl_3_) δ = 164.8 (C), 51.8 (CH), 32.8 (2CH_2_), 29.4 (CH_2_), 23.6 (2CH_2_).

2-Bromo-*N*-(2-bromo-5-fluorophenyl)acetamide (**14**)

The crude product was purified by column chromatography (silica gel, dichloromethane/petroleum ether 5/5). Yield 25% (280 mg). White solid. Mp. 99–100 °C. 1H NMR (400 MHz, CDCl_3_) δ 8.86 (s, 1H, NH), 8.24–8.21 (m, 1H, CH), 7.53–7.50 (m, 1H, CH), 6.81–6.76 (m, 1H, CH), 4.08 (s, 2H, CH_2_). 13C NMR (100 MHz, CDCl_3_) δ 163.7 (C), 162.1 (d, 1*J*_C-F_ = 246.3 Hz, C), 136.3 (d, 3*J*_C-F_ = 11.6 Hz, C), 133.1 (d, 3*J*_C-F_ = 9.5 Hz, CH), 113.0 (d, 2*J*_C-F_ = 23.2 Hz, CH), 108.9 (d, 2*J*_C-F_ = 29.1 Hz, CH), 107.7 (d, 4*J*_C-F_ = 3.6 Hz, C), 29.7 (CH_2_).

#### 3.1.4. General Procedure for the Synthesis of 2-Nitropyrrole Derivatives

To a solution of 2-nitropyrrole (0.1 g, 0.9 mmol, 1 eq) in ethanol (5 mL) was added K_2_CO_3_ (0.138 g, 0.99 mmol, 1.1 eq), and the corresponding bromoacetamide compound (1.1 eq). The reaction mixture was stirred and heated at 70 °C for 3 h. After cooling, the solution was poured into a bath of ice. A precipitate appeared and was filtered, washed with water (3 × 100 mL), petroleum ether (3 × 100 mL), and dried in a vacuum drying oven (desiccator cabinet).

*N*-Benzyl-2-(2-nitro-1*H*-pyrrol-1-yl)acetamide (**3**)

Yield 73% (113 mg). White solid. Mp 174–175 °C. 1H NMR (400 MHz, DMSO-d6) δ 8.70 (s, 1H, NH), 7.33–7.25 (m, 7H, 7CH), 6.28 (dd, 3*J*_H-H_ = 4.4 Hz, 4*J*_H-H_ =2.7 Hz, 1H, CH), 5.09 (s, 2H, CH_2_), 4.32 (d, 3*J*_H-H_ = 5.9 Hz, 2H, CH_2_). 13C NMR (100 MHz, DMSO-d6) δ 167.1 (C), 139.5 (C), 137.6 (C), 132.9 (CH), 128.7 (2CH), 127.7 (2CH), 127.3 (CH), 114.8 (CH), 109 (CH), 52.6 (CH_2_), 42.7 (CH_2_). LC-MS (ESI+) Tr 4.12 min, *m*/*z* [M + H]^+^ 260.18. MW: 259.10 g·mol^−1^. HRMS: *m*/*z* [M + H]^+^ calcd for [C_13_H_13_N_3_O_3_]^+^: 260.1030; found: 260.1029.

2-(2-Nitro-1*H*-pyrrol-1-yl)-*N*-phenylacetamide (**4**)

Yield 78% (170 mg). White solid. Mp 201–202 °C. 1H NMR (400 MHz, DMSO-d6) δ 10.4 (s, 1H, NH), 7.57–7.55 (m, 2H, 2CH), 7.36–7.28 (m, 4H, 4CH), 7.06 (t, 3*J*_H-H_ = 7.4 Hz, 1H, CH), 6.32 (dd, 3*J*_H-H_ = 4.4 Hz, 4*J*_H-H_ = 2.7 Hz, 1H, CH), 5.24 (s, 2H, CH_2_). 13C NMR (100 MHz, DMSO-d6) δ 165.3 (C), 138.7 (C), 137.0 (C), 132.5 (CH), 128.9 (2CH), 123.5 (CH), 119.0 (2CH), 114.4 (CH), 108.6 (CH), 52.8 (CH_2_). LC-MS (ESI+) Tr 4.08 min, *m*/*z* [M + H]^+^ 246.21. MW: 245.08 g·mol^−1^. HRMS: *m*/*z* [M + H]^+^ calcd for [C_12_H_11_N_3_O_3_]^+^: 246.0873; found: 246.0873.

2-(2-Nitro-1*H*-pyrrol-1-yl)-*N*-[3-(trifluoromethyl)benzyl]acetamide (**15**)

Yield 60% (172 mg). White solid. Mp 161–162 °C. 1H NMR (400 MHz, DMSO-d6) δ 8.81 (s, 1H, NH), 7.64–7.57 (m, 4H, 4CH), 7.33–7.30 (m, 1H, CH), 7.25 (dd, 3*J*_H-H_ = 4.1 Hz, 4*J*_H-H_ = 1.9 Hz 1H, CH), 6.28 (dd, 3*J*_H-H_ = 3.8 Hz, 4*J*_H-H_ = 2.7 Hz, 1H, CH), 5.11 (s, 2H, CH_2_), 4.42 (d, 3*J*_H-H_ = 5.8 Hz, 2H, CH_2_). 13C NMR (100 MHz, DMSO-d6) δ 167.0 (C), 140.7 (C), 137.1 (C), 132.4 (CH), 131.2 (CH), 129.3 (CH), 129.1 (q, 2*J*_C-F_ = 31.8 Hz, C), 124.2 (q, 1*J*_C-F_ = 272.4 Hz, C), 123.6 (q, 3*J*_C-F_ = 3,8 Hz, 2CH), 114.3 (CH), 108.5 (CH), 52.2 (CH_2_), 41.7 (CH_2_). LC-MS (ESI+) Tr 5.07 min, *m*/*z* [M + H]^+^ 328.09. MW: 327.08g·mol^−1^. HRMS: *m*/*z* [M + H]^+^ calcd for [C_14_H_12_F_3_N_3_O_3_]^+^: 328.0904; found: 328.0901.

*N*-(4-Bromophenyl)-2-(2-nitro-1*H*-pyrrol-1-yl)acetamide (**16**)

Yield 86% (291 mg). White solid. Mp 229–230 °C. 1H NMR (400 MHz, DMSO-d6) δ 10.52 (s, 1H, NH), 7.55–7.52 (m, 4H, 4CH), 7.35–7.34 (m, 1H, CH), 7.34–7.30 (m, 1H, CH), 6.32 (dd, 3*J*_H-H_ = 3.8 Hz, 4*J*_H-H_ = 2.8 Hz, 1H, CH), 5.24 (s, 2H, CH_2_). 13C NMR (100 MHz, DMSO-d6) δ 165.6 (C), 138.0 (C), 137.0 (C), 132.5 (CH), 131.7 (2CH), 121.0 (2CH), 115.1 (C), 114.1 (CH), 108.7 (CH), 52.8 (CH_2_). LC-MS (ESI+) Tr 5.05 min, *m*/*z* [M + H]^+^ 323.98. MW: 322.99 g·mol^−1^. HRMS: *m*/*z* [M + H]+ calcd for [C_12_H_10_BrN_3_O_3_]^+^: 323.9978; found: 323.9977.

*N*-(2-Chlorophenyl)-2-(2-nitro-1*H*-pyrrol-1-yl)acetamide (**17**)

Yield 96% (251 mg). White solid. Mp 178–179 °C. 1H NMR (400 MHz, DMSO-d6) δ 9.98 (s, 1H, NH), 7.68 (dd, 3*J*_H-H_ = 8.1 Hz, 4*J*_H-H_ = 1.4 Hz, 1H, CH), 7.51 (dd, 3*J*_H-H_ = 8.1 Hz, 4*J*_H-H_ = 1.2 Hz, 1H, CH), 7.39–7.37 (m, 1H, CH), 7.34–7.31 (m, 1H, CH), 7.28 (dd, 3*J*_H-H_ = 4.4 Hz, 4*J*_H-H_ = 2.0 Hz, 1H, CH), 7.22–7.18 (m, 1H, CH), 6.31 (dd, 3*J*_H-H_ = 4.4 Hz, 4*J*_H-H_ = 2.7 Hz, 1H, CH), 5.33 (s, 2H, CH_2_). 13C NMR (100 MHz, DMSO-d6) δ 166.1 (C), 137.0 (C), 134.4 (C), 132.4 (CH), 129.6 (CH), 127.5 (CH), 126.5 (CH), 126.2 (C), 125.9 (CH), 114.4 (CH), 108.6 (CH), 52.6 (CH_2_). LC-MS (ESI+) Tr 4.44 min, *m*/*z* [M + H]^+^ 280.02. MW: 279.04 g·mol^−1^. HRMS: *m*/*z* [M + H]^+^ calcd for [C_12_H_10_ClN_3_O_3_]^+^: 280.0483; found: 280.0483.

2-(2-Nitro-1*H*-pyrrol-1-yl)-*N*-[3-(trifluoromethyl)phenyl]acetamide (**18**)

Yield 81% (217 mg). White solid. Mp 202–203 °C. 1H NMR (400 MHz, DMSO-d6) δ 10.7 (s, 1H, NH), 8.06 (s, 1H, CH), 7.75–7.73 (m, 1H, CH), 7.55–7.59 (m, 1H, CH), 7.44–7.42 (m, 1H, CH), 7.37–7.35 (m, 1H, CH), 7.30 (dd, 3*J*_H-H_ = 4.3 Hz, 4*J*_H-H_ = 2.1 Hz, 1H, CH), 6.33 (dd, 3*J*_H-H_ = 4.4 Hz, 4*J*_H-H_ = 2.7 Hz, 1H, CH), 5.27 (s, 2H, CH_2_). 13C NMR (100 MHz, DMSO-d6) δ 166.1 (C), 139.4 (C), 137.0 (C), 132.5 (CH), 130.2 (CH), 129.5 (q, *2J_C-F_* = 32 Hz, C), 124.0 (q, *1J_C-F_* = 271.0 Hz, C), 122.6 (CH), 119.8 (q, *3J_C-F_* = 4.4 Hz, CH), 115.0 (q, *3J_C-F_* = 3.6 Hz, CH), 114.4 (CH), 108.7 (CH), 52.8 (CH_2_). LC-MS (ESI+) Tr 5.21 min, *m*/*z* [M + H]^+^ 314.11. MW: 313.07 g·mol^−1^. HRMS: *m*/*z* [M + H]^+^ calcd for [C_13_H_10_F_3_N_3_O_3_]^+^: 314.0747; found: 314.0745.

*N*-(3-Methoxyphenyl)-2-(2-nitro-1*H*-pyrrol-1-yl)acetamide (**19**)

Yield 88% (217 mg). White solid. Mp 192–193 °C. 1H NMR (400 MHz, DMSO-d6) δ 10.39 (s, 1H, NH), 7.35–7.21 (m, 4H, 4CH), 7.08 (s, 1H, CH), 6.64 (s, 1H, CH), 6.32 (s, 1H, CH), 5.23 (s, 2H, CH_2_), 3.72 (s, 3H, CH_3_). 13C NMR (100 MHz, DMSO-d6) δ 165.4 (C), 159.6 (C), 139.8 (C), 137.0 (C), 132.5 (CH), 129.7 (CH), 114.4 (CH), 111.2 (CH), 109.0 (CH), 108.6 (CH), 104.7 (CH), 54.9 (CH_3_), 52.8 (CH_2_). LC-MS (ESI+) Tr 4.32 min, *m*/*z* [M + H]^+^ 276.11. MW: 275.09 g·mol^−1^. HRMS: *m*/*z* [M + H]^+^ calcd for [C_13_H_13_N_3_O_4_]^+^: 276.0979; found: 276.0979.

*N*-Isopropyl-2-(2-nitro-1*H*-pyrrol-1-yl)acetamide (**20**)

Yield 30% (57 mg). White solid. Mp 190–191 °C. 1H NMR (400 MHz, DMSO-d6) δ 8.09 (d, 3*J*_H-H_ = 7.5 Hz, 1H, NH), 7.27 (t, 3*J*_H-H_ = 2.4 Hz, 1H, CH), 7.22 (dd, 3*J*_H-H_ = 4.2 Hz, 4*J*_H-H_ = 2.1 Hz, 1H, CH), 6.25 (dd, 3*J*_H-H_ = 4.2 Hz, 4*J*_H-H_ = 2.7 Hz, 1H, CH), 4.95 (s, 2H, CH_2_), 3.86–3.78 (m, 1H, CH), 1.07 (d, 3*J*_H-H_ = 6.6 Hz, 6H, 2CH_3_). 13C NMR (100 MHz, DMSO-d6) δ 165.3 (C), 137.1 (C), 132.3 (CH), 114.2 (CH), 108.3 (CH), 52.1 (CH), 40.7 (CH_2_), 22.4 (2CH_3_). LC-MS (ESI+) Tr 3.16 min, *m*/*z* [M + H]^+^ 212.34. MW: 211.10 g·mol^−1^. HRMS: *m*/*z* [M + H]^+^ calcd for [C_9_H_13_N_3_O_3_]^+^: 212.1030; found: 212.1029.

*N*-Cyclopentyl-2-(2-nitro-1*H*-pyrrol-1-yl)acetamide (**21**)

Yield 59% (125 mg). White solid. Mp 190–191 °C. 1H NMR (400 MHz, DMSO-d6) δ 8.17 (d, 3*J*_H-H_ = 7.3 Hz, 1H, NH), 7.28 (t, 3*J*_H-H_ = 2.2 Hz, 1H, CH), 7.23 (dd, 3*J*_H-H_ = 4.2 Hz, 4*J*_H-H_ = 2.1 Hz, 1H, CH), 6.26 (dd, 3*J*_H-H_ = 4.4 Hz, 4*J*_H-H_ = 2.7 Hz, 1H, CH), 4.96 (s, 2H, CH_2_), 4.01–3.96 (m, 1H, CH), 1.82–1.76 (m, 2H, CH_2_), 1.67–1.64 (m, 2H, CH_2_), 1.53–1.49 (m, 2H, CH_2_), 1.44–1.37 (m, 2H, CH_2_). 13C NMR (100 MHz, DMSO-d6) δ 165.7 (C), 137.1 (C), 132.3 (CH), 114.2 (CH), 108.3 (CH), 52.1 (CH), 50.5 (CH_2_), 32.3 (2 CH_2_), 23.4 (2 CH_2_). LC-MS (ESI+) Tr 3.85 min, *m*/*z* [M + H]^+^ 238.24. MW: 237.11 g·mol^−1^. HRMS: *m*/*z* [M + H]^+^ calcd for [C_11_H_15_N_3_O_3_]^+^: 238.1186; found: 238.1191.

*N*-(2-Bromo-5-fluorophenyl)-2-(2-nitro-1*H*-pyrrol-1-yl)acetamide (**22**)

Yield 77% (123 mg). Yellow solid. Mp 185–186 °C. 1H NMR (400 MHz, DMSO-d6) δ 9.98 (s, 1H, NH), 7.75–7.71 (m, 1H, CH), 7.56–7.52 (m, 1H, CH), 7.07–7.02 (m, 1H, CH), 7.39 (t, 3*J*_H-H_ = 2.3 Hz, 1H, CH), 7.30 (dd, 3*J*_H-H_ = 4.4 Hz, 4*J*_H-H_ = 2.0 Hz, 1H, CH), 6.33 (dd, 3*J*_H-H_ = 4.4 Hz, 4*J*_H-H_ = 2.7 Hz, 1H, CH), 5.35 (s, 2H, CH_2_). 13C NMR (100 MHz, DMSO-d6) δ 166.4 (C), 161.0 (d, 1*J*_C-F_ = 241.2 Hz, C), 137.2 (C), 137.0 (d, 3*J*_C-F_ = 9.4 Hz, C), 134.0 (d, 3*J*_C-F_ = 9.4 Hz, CH), 132.4 (CH), 114.5 (CH), 113.8 (d, 2*J*_C-F_ = 22.5 Hz, CH), 112.6 (d, 2*J*_C-F_ = 26.1 Hz, CH), 110.9 (C), 108.7 (CH), 52.7 (CH_2_). LC-MS (ESI+) Tr 4.84 min, *m*/*z* [M + H]^+^ 342.0. MW: 340.98 g·mol^−1^. HRMS: *m*/*z* [M + H]^+^ calcd for [C_12_H_9_BrFN_3_O_3_]^+^: 341.9884; found: 341.9877.

#### 3.1.5. Crystal Data for 2-Nitropyrrole Starting Product 1

Crystal Data for Compound 1 C_4_H_4_N_2_O_2_, M = 112.09, a = 7.7417(3) Å, b = 10.1820(5) Å, c = 21.2170(6) Å, α = 89.654(3)◦, β = 89.210(3)◦, γ = 69.065(4)◦, V = 1507.42(11) Å3, T = 242 K, space group P1 Z = 12, 5673 independent reflections (Rint = 5673). The final R1 values were 0.0721 (I > 2σ(I)). The final wR(F 2) values were 0.1919 (I > 2σ(I)). The goodness of fit on F 2 was 1.159. CCDC 2132900 contains the supplementary crystallographic data for this paper. These data can be obtained free of charge at www.cdcc.cam.ac.uk/data_request/cif (accessed on 24 February 2022) of from the Cambridge Crystallographic Data Centre, 12, Union Road, Cambridge CB2 1EZ, UK; Fax: + 44 (1223) 336033; email: deposit@ccdc.cam.ac.uk.

### 3.2. Toxicology

The toxicity of compounds towards Chinese hamster ovary cells (CHO-K1, ATCC CCL61) was evaluated by the NRU toxicity test, based on the concentration-dependent reduction of the uptake of the vital dye neutral red when measured 24 h after chemical treatment. Cells were grown in McCoy’s medium supplemented with Penicillin 100 IU/mL and streptomycin 100 μg/mL, and 10% of inactivated calf serum, and incubated at 37 °C in CO_2_ 5% atmosphere. They were seeded into two 96-well tissue culture plates (0.1 mL per well), at a concentration of 1 × 105 cells/mL, and incubated at 37 °C (5% CO_2_) for 24 h, until confluent. The culture medium was decanted and replaced by 100 µL of fresh medium containing the appropriate concentrations of the compounds (eight different concentrations in triplicate), then cells were incubated at 37 °C (5% CO_2_) in the dark for 24 h. At the end of the incubation period, cells were washed, placed into neutral red medium (50 μg/mL neutral red in complete medium). This vital dye is a weak cationic dye that penetrates membranes of living cells by non-diffusion and accumulates intracellularly in lysosomes. On the contrary, alterations of the cell surface or metabolism result in a decreased uptake and binding of NR in non-viable cells. Cells were incubated for 3 h at 37°C, 5% CO_2_, and then the culture medium was removed, and cells were washed three times with 0.2 mL of PBS to remove excessive dye. Destaining solution (50% ethanol, 1% acetic acid, 49% distilled water; 50µL per well) was added into the wells, and the plates were shaken for 15–20 min at room temperature in the dark. The degree of cell viability was measured by a fluorescence–luminescence reader. The optical density (OD) of each well was read at 540 nm. The results obtained for wells treated with the test compounds were compared to those of untreated control wells (100% viability) and converted to percentage values. The mean OD value of blank wells (containing only neutral red desorbed solution) was subtracted from the mean OD value of three treated wells (dilutions of the test material, positive control or HBSS). The percentages of cell viability were calculated as:Viability (%) = (Mean OD of test wells-mean OD of blanks) × 100 / (Mean OD of negative control − mean OD of blanks)

The concentration of the test substances causing a 50% release of neutral red as compared to the control culture was calculated by non-linear regression analysis using software Phototox Version 2.0.

### 3.3. T. cruzi Intracellular Assay 

Vero cells (ECCAC 84113001) were screened for mycoplasma infection and maintained in MEM medium supplemented with Glutamax (Life Technologies) and 10% (*v/v*) foetal calf serum (FCS) at 37 °C in the presence of 5% CO2. *T. cruzi* parasites (Silvio X10/7 A1), a clonal line kindly provided by Susan Wyllie and Prof. Alan Fairlamb were maintained in Vero cells. On a weekly basis, emerged trypomastigotes were used to infect a new Vero monolayer at the multiplicity of infection (MOI) 1.5 [32].

For the primary intracellular assay, the infection conditions were chosen so that no trypomastigote egress occurs during the compound incubation time. First, Vero cells were infected overnight with tissue culture-derived *T. cruzi* trypomastigotes in T225 tissue culture flasks (MOI 5).

Next, any remaining free trypomastigotes were washed away with serum-free MEM, and the infected Vero cells were harvested by trypsinisation. The infected Vero cells were then plated into 384-well plates containing the compounds to be tested, at 4000 cells per well in MEM media with 1% FCS. After 96h incubation at 37 °C in the presence of 5% CO2, the plates were fixed with 4% formaldehyde for 20 min at room temperature and stained with 5μg/mL Hoechst 33342. The plates were imaged on a Perkin Elmer Operetta high-content imaging system using a 20× objective. Images were analysed using the Columbus system (Perkin Elmer). The algorithm for the primary assay first identified the Vero nuclei followed by the demarcation of the cytoplasm and identification of intracellular amastigotes. This algorithm reported the mean number of parasites per Vero cell and the total number of Vero cells. The data analysis was as previously described [33].

The robust z-factor was calculated with the following formula:
$$1-\frac{\left(3\times \left(1.4826\times \mathrm{MAD}\left[0\%\;\mathrm{inhibition}\;\left(\mathrm{raw}\;\mathrm{data}\right)\right]\right)\right)+\left(3\times \left(1.4826\times \mathrm{MAD}\left[100\%\;\mathrm{inhibition}\;\left(\mathrm{raw}\;\mathrm{data}\right)\right]\right)\right)}{\mathrm{MEDIAN}\;\left[0\%\;\mathrm{inhibition}\;\left(\mathrm{raw}\;\mathrm{data}\right)\right]-\mathrm{MEDIAN}\;\left[100\%\;\mathrm{inhibition}\;\left(\mathrm{raw}\;\mathrm{data}\right)\right]}$$
with MAD = Mean Absolute Deviation. Hits from the primary screen were selected based on the following criteria: *T. cruzi* activity (percent inhibition) > (median *T. cruzi* percent inhibition + 3 × robust Standard Deviation) and Vero activity (percent inhibition) < (median percent Vero inhibition + 3 × robust Standard Deviation).

## 4. Conclusions

A series of 10 novel 1-functionalized 2-nitropyrrole compounds were designed in accordance with Lipinski’s rule of five, and synthesized with good yields. The convergent synthesis methodology allowed us to introduce in 1-position of the 2-nitropyrrole an acetamide side-chain close to that found in benznidazole by overcoming the pyrrole polymerization problem. The cytotoxicity evaluation on CHO cells showed no significant cytotoxicity, except for compound **3** (CC_50_ < 30 µM). Compound **18** appeared to show that activity against *T. cruzi* intracellular amastigotes forms (EC_50_ = 3.6 ± 1.8 µM) and good selectivity over the Vero host cells. Unfortunately, this compound **18** showed an insufficient maximum effect compared to the reference drug (nifurtimox) to ensure the further development of antichagasic compounds.

## Data Availability

Not applicable.

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
