# Peer review of "Synthesis and Anti-Trypanosoma cruzi Biological Evaluation of Novel 2-Nitropyrrole Derivatives"

_molecules, 2022, doi:10.3390/molecules27072163_

Round 1

Reviewer 1 Report

This paper reporting on synthesis of 2-nitropyrroles and their biological evaluation for anti-trypanosoma cruzi activity can be considered of low originality not only due to a highly limited structural versatility of the presented drug candidates with simple structures. With the intention of extending SAR, further relevant five-membered heterocycles, e.g. 5-nitro-1H-pyrazole should have been explored in the design, synthesis and evaluation of drug candidates. The other main problem with this research is that 18 was the exclusive model compound subjected to Trypanosoma cruzi assay. On the other hand, more prolonged treatments were also neglected. Finally, it is not clear why Nifurtimox was used as reference instead of the structurally related Benznidazole, the obviously reasonable choice.

Author Response

Dear reviewer,

Structures of the presented drug candidates are simple, but this work shows anti-T. cruzi evaluation of 2-nitropyrrole derivatives not yet described in the literature. It provides a complement in the study of the antitrypanosomatid properties of nitroheterocycles.

Presentation of the biological results was too succinct, and some errors led to misunderstandings. We corrected compound 13 to compound 3 in Table 4 and the biological part was expanded with complementary data:

  • cruzi EC50 for all compounds tested,
  • cruzi EC50 for Nifurtimox reference drug in Table 4
  • Vero cells EC50 for all compounds tested,
  • Dose-response curve data of compound 18 and nifurtimox in the manuscript,
  • Dose-response curve data of the other compounds tested in the supplementary material,
  • Complete revision of the discussion: “These new derivatives were evaluated in vitro against the intracellular cruzi strain X10/7-Silvio by determining their 50% efficacy concentration (EC50) and were compared to the reference anti-T. cruzi drug nifurtimox. The in vitro 50% cytotoxic concentrations (CC50) were assessed on the CHO cell line and compared to a cytotoxic reference drug, doxorubicin. In vitro cytotoxicity against vero host cells was also assessed (vero cells EC50). The results are presented in Table 4.

Regarding the effect on cell viability of the CHO cells, except for one compound (3), the other derivatives showed good cytotoxicity values ranging from 155.6 to 321.1 µM compared to the reference drug, doxorubicin (CC50 CHO = 4.6 µM). Three compounds showed no cytotoxicity against vero host cells at concentrations up to 50 µM (17-19).

Of the 10 synthesized molecules, three showed poor DMSO solubility, limiting their in vitro anti-T. cruzi evaluation (20-22). Three compounds showed T. cruzi EC50 < 10 µM (4, 16 and 18). Only compound 18 exhibited activity against the parasite (EC50 = 3.6 ± 1.8 µM) without toxicity against host cells at concentrations up to 50 µM. However, the maximum effect of the latter compound (Figure 4) compared to the nifurtimox control (Figure 5) was relatively low (79% and 71% in two independent replicates), indicating that a substantial number of parasites survive 96h treatment. Whether longer duration treatments may eliminate all parasites remains to be explored.

Structure-Activity Relationship (SAR) can be suggested from these biological results. The presence of a benzyl group seems to be unfavourable for anti-T. cruzi activity. Indeed, compounds 3 and 15 with respectively benzyl and meta-EWG 3-CF3 benzyl group exhibit no activity against T. cruzi (EC50 = 34.5 µM and 41.0 µM) whereas their analogue with phenyl (4) and meta-EWG 3-CF3 phenyl group (18) exhibit activity (EC50 < 10 µM). The meta-EWG 3-CF3 phenyl group gave the best anti-T. cruzi activity (18) while the meta-electro-donating OCH3 phenyl group (19) gave the lowest anti-T. cruzi activity (EC50 = 50.0 µM). An electron-withdrawing (EWG) group in the meta-position of the phenyl seems to be important for anti-T. cruzi activity.”

Benznidazole and nifurtimox compounds are used clinically. Our biologist collaborator used nifurtimox for in vitro assays due to its slightly higher potency.

Reviewer 2 Report

In this manuscript, Mathias et al report the synthesis of 2-nitropyrrole derivatives and their activity against T. cruzi in vitro. I do not have medicinal chemistry expertise and thus will not comment on the chemistry sections of this manuscript. With regards to Chagas disease literature and antiparasitic activity assays, major concerns are as follows:

  1. Dose-response curve data should be provided for EC50 and CC50 calculations. This is especially important since authors are comparing the maximum effect between their compound and nifurtimox.
  2. EC50 and dose-response curve for nifurtimox reference should be provided to enable comparison to the novel compound.
  3. Only comparison to nifurtimox is mentioned in the results section, but conclusion and abstract compare compound 18 to both nifurtimox and benznidazole. If both reference drugs were tested, data for both should be provided. If only one was tested, text should be updated accordingly.
  4. Rationale for only testing compound 18 against cruzi should be provided. The value of this paper to the field and the ability to generate SAR conclusions would be much enhanced if all the non-cytotoxic derivatives were tested.
  5. Methods describe a primary screen with hit criteria, but this is inconsistent with the results presented in the manuscript, where only EC50 for a single compound is reported.
  6. Line 403: 1.105 cells/ml. Do the authors mean 1x10 to the power of 5? 1.105 per mL seems very low.
  7. The specific cruzi strain used for assays should be listed.
  8. Line 466 lists no funding, but then funding support is listed in the acknowledgements. Please correct.

Minor comments:

  1. Line 17, Trypanosoma should be capitalized.
  2. Lines 42-43: this statement should be tempered. Activity in early chronic stage has been observed, but not once late-stage, advanced heart damage is evident.
  3. Lines 152-153: this statement should be tempered. Not all drugs comply with Lipinski’s rule of five, especially in the antimicrobial field. See PMCID: PMC5473536 for example.
  4. Lines 157-158: “There are not parameters out of ranger, so we didn’t identify alert of poor absorption or permeability for the oral route of administration.” should be corrected to “There are no parameters out of range, so we didn’t identify any alerts of poor absorption or permeability for the oral route of administration.”
  5. Line 412 typo: “Distaining” should read “destaining”

Author Response

Dear Reviewer,

In response to your comments:

1. Dose-response curve data for compound 18 against T. cruzi X10/7 intracellular amastigote and inhibition of vero host cell growth were added in the manuscript. Dose-response curve of the other compounds tested were included in supplementary data.

2. T. cruzi EC50 for Nifurtimox was added in Table 4. Dose-response curve data for nifurtimox against T. cruzi X10/7 intracellular amastigote and inhibition of vero host cell growth were added in the supplementary data.

3. Only nifurtimox was used as reference drug. The text was updated in abstract and conclusion.

4. Seven compounds were tested but only compound 18 showed IC50 < 10 µM and good selectivity over the vero host cells. The biological part was expanded with complementary data:

  • cruzi EC50 for all compounds tested,
  • Vero cells EC50 for all compounds tested,
  • Dose-response curve data of compound 18 and nifurtimox in the manuscript,
  • Dose-response curve data of the other compound tested in the supplementary material,
  • Complete revision of the discussion: “These new derivatives were evaluated in vitro against the intracellular cruzi strain X10/7-Silvio by determining their 50% efficacy concentration (EC50) and were compared to the reference anti-T. cruzi drug nifurtimox. The in vitro 50% cytotoxic concentrations (CC50) were assessed on the CHO cell line and compared to a cytotoxic reference drug, doxorubicin. In vitro cytotoxicity against vero host cells was also assessed (vero cells EC50). The results are presented in Table 4.

Regarding the effect on cell viability of the CHO cells, except for one compound (3), the other derivatives showed good cytotoxicity values ranging from 155.6 to 321.1 µM compared to the reference drug, doxorubicin (CC50 CHO = 4.6 µM). Three compounds showed no cytotoxicity against vero host cells at concentrations up to 50 µM (17-19).

Of the 10 synthesized molecules, three showed poor DMSO solubility, limiting their in vitro anti-T. cruzi evaluation (20-22). Three compounds showed T. cruzi EC50 < 10 µM (4, 16 and 18). Only compound 18 exhibited activity against the parasite (EC50 = 3.6 ± 1.8 µM) without toxicity against host cells at concentrations up to 50 µM. However, the maximum effect of the latter compound (Figure 4) compared to the nifurtimox control (Figure 5) was relatively low (79% and 71% in two independent replicates), indicating that a substantial number of parasites survive 96h treatment. Whether longer duration treatments may eliminate all parasites remains to be explored.

Structure-Activity Relationship (SAR) can be suggested from these biological results. The presence of a benzyl group seems to be unfavourable for anti-T. cruzi activity. Indeed, compounds 3 and 15 with respectively benzyl and meta-EWG 3-CF3 benzyl group exhibit no activity against T. cruzi (EC50 = 34.5 µM and 41.0 µM) whereas their analogue with phenyl (4) and meta-EWG 3-CF3 phenyl group (18) exhibit activity (EC50 < 10 µM). The meta-EWG 3-CF3 phenyl group gave the best anti-T. cruzi activity (18) while the meta-electro-donating OCH3 phenyl group (19) gave the lowest anti-T. cruzi activity (EC50 = 50.0 µM). An electron-withdrawing (EWG) group in the meta-position of the phenyl seems to be important for anti-T. cruzi activity.”

5. To enhance the value of this paper as requested in Q4, we have incremented the biological paragraph with the results of all the molecules tested, even if only molecule 18 exhibit activity against T. cruzi and good selectivity over the host cells. Discussion of the biological part was expanded, and Structure-Activity Relationship were established.

6. Line 403: Yes we meant 1x10 to the power of 5. This typology was corrected.

7. The specific cruzi strain used X10/7-Silvio was added in the experimental section part 3.3 T. cruzi intracellular assay.

8. Line 466: Funding was added

Minor comments:

  1. Line 17, Trypanosoma was capitalized.
  2. Lines 42-43: This statement was corrected to “These two compounds represent an important part of the therapeutic arsenal in the management of Chagas disease. However, they are active mainly in the acute and early chronic stage of the disease, but not once late-stage”.
  3. Lines 152-153: This statement was corrected to “Compliance with Lipinski’s rule of five is reassuring, but we must keep in mind that not all drugs fully comply with this rule, especially in the antimicrobial field” and the suggested reference was added.
  4. Lines 157-158: Sentence “There are not parameters out of ranger, so we didn’t identify alert of poor absorption or permeability for the oral route of administration.” was corrected to “There are no parameters out of range, so we didn’t identify any alerts of poor absorption or permeability for the oral route of administration.”
  5. Line 412 typo: “Distaining” was corrected to “destaining”

Reviewer 3 Report

The authors report on the synthesis of 2-nitropyrrole derivatives and their anti-trypanosoma cruzi activity.  In my opinion, this manuscript does not provide novelty in terms of the design of the compounds and their synthesis. Additionally, the activity of these compounds is very low and they show high cytotoxicity. The synthetic methodologies used were reported before and are straightforward. Unfortunately, I can not recommend acceptance of this manuscript in high-impact journal like Molecule. Perhaps the authors can consider Synthetic Communications.

Additional comments:

Abstract: change “disease related to T. cruzi” to “disease caused by T. cruzi”

Line 52: change in to on.

Lines 58-62. Need to be revised.

Line 85: change “dinitrate pyrrole” to “dinitro pyrrole”

Line 100: change “benzyl or phenyl amine” to bezylamine and aniline”

Line 142: something is wrong here since compound 13 is not in the table nor cc50 30.

In my view, the cytotoxicity of compounds 3-4, 15-22 is way higher than doxorubicin and on average 50 times higher. Authors can not conclude that the cytotoxicity is acceptable.

Author Response

Dear Reviewer,

This manuscript presents anti-T. cruzi evaluation of 2-nitropyrrole derivatives not yet described in the literature. It provides a complement in the study of the anti-T. cruzi properties of nitroheterocycle compounds.

Presentation of the biological results was too succinct and some errors led to misunderstandings. We corrected compound 13 to compound 3 in Table 4 and the biological part was expanded with complementary data:

  • cruzi EC50 for all compounds tested,
  • cruzi EC50 for Nifurtimox reference drug in Table 4
  • Vero cells EC50 for all compounds tested,
  • Dose-response curve data of compound 18 and nifurtimox in the manuscript,
  • Dose-response curve data of the other compound tested in the supplementary material,
  • Complete revision of the discussion: “These new derivatives were evaluated in vitro against the intracellular cruzi strain X10/7-Silvio by determining their 50% efficacy concentration (EC50) and were compared to the reference anti-T. cruzi drug nifurtimox. The in vitro 50% cytotoxic concentrations (CC50) were assessed on the CHO cell line and compared to a cytotoxic reference drug, doxorubicin. In vitro cytotoxicity against vero host cells was also assessed (vero cells EC50). The results are presented in Table 4. Regarding the effect on cell viability of the CHO cells, except for one compound (3), the other derivatives showed good cytotoxicity values ranging from 155.63 to 321.12 µM compared to the reference drug, doxorubicin (CC50 CHO = 4.65 µM). Three compounds showed no cytotoxicity against vero host cells at concentrations up to 50 µM (17-19). Of the 10 synthesized molecules, three showed poor DMSO solubility, limiting their in vitro anti-T. cruzi evaluation (20-22). Three compounds showed T. cruzi EC50 < 10 µM (4, 16 and 18). Only compound 18 exhibited activity against the parasite (EC50 = 3.6 ± 1.8 µM) without toxicity against host cells at concentrations up to 50 µM. However, the maximum effect of the latter compound (Figure 4) compared to the nifurtimox control (Figure 5) was relatively low (79% and 71% in two independent replicates), indicating that a substantial number of parasites survive 96h treatment. Whether longer duration treatments may eliminate all parasites remains to be explored. Structure-Activity Relationship (SAR) can be suggested from these biological results. The presence of a benzyl group seems to be unfavourable for anti-T. cruzi activity. Indeed, compounds 3 and 15 with respectively benzyl and meta-EWG 3-CF3 benzyl group exhibit no activity against T. cruzi (EC50 = 34.5 µM and 41.0 µM) whereas their analogue with phenyl (4) and meta-EWG 3-CF3 phenyl group (18) exhibit activity (EC50 < 10 µM). The meta-EWG 3-CF3 phenyl group gave the best anti-T. cruzi activity (18) while the meta-electro-donating OCH3 phenyl group (19) gave the lowest anti-T. cruzi activity (EC50 = 50.0 µM). An electron-withdrawing (EWG) group in the meta-position of the phenyl seems to be important for anti-T. cruzi activity.”

On the other hand, the synthesis methodology required optimization work. Indeed, pyrrole alkylation was not easy and led us to use a convergent methodology whereas for benznidazole the alkylation is done directly on the imidazole scaffold and doesn’t require the synthesis of reactional intermediates.

Additional comments:

Abstract: “disease related to T. cruzi” was corrected to “disease caused by T. cruzi

Line 52: “in” was corrected to “on”

Lines 58-62. This paragraph was revised to “The aim of this work was to synthesize 2-nitropyrrole derivatives functionalized at the N1-position by a chain structurally close to that found in benznidazole (Figure 2). Several nitro-drug are known to act as a prodrug bioactivated by T. cruzi nitro-reductases NTR type I [14]-[15] capable of exerting cytotoxic effects by DNA damage. We will investigate wether the synthesized 2-nitropyrrole derivatives exhibit anti-T. cruzi activity.”

Line 85: “dinitrate pyrrole” was corrected to “dinitro pyrrole”

Line 100: “benzyl or phenyl amine” was corrected to benzylamine and aniline”

Line 142: 13 was corrected to compound 3 throughout the manuscript.

To answer your question about cytotoxicity, we confirm that compounds 4, 15-22 have a lower cytotoxicity than doxorubicin (50 times lower) in CHO cells. Indeed, when the CC50 of the compound tested is higher than the CC50 of the referent cytotoxicity drug doxorubicin, the compound is less cytotoxic.

Cytotoxicity paragraph was revised to: “The in vitro 50% cytotoxic concentrations (CC50) were assessed on the CHO cell line and compared to a cytotoxic reference drug, doxorubicin. The in vitro cytotoxicity against the vero host cells was also evaluated (vero cells EC50). The results are presented in Table 4. Regarding the effect on cell viability of the CHO cells, except for one compound (3), the other derivatives showed good cytotoxicity values ranging from 155.6 to 321.1 µM in comparison with reference drug doxorubicin (CC50 CHO = 4.6 µM). Three compounds showed no cytotoxicity against the vero host cells at concentrations up to 50 µM (17-19).”

We added EC50 values in the vero host cells and dose response curves inhibition of vero host cell growth (Figure 4 and Figure 5). The dose-response curve date showed the negative inhibition of vero cells at concentrations that inhibit parasite growth.

Reviewer 4 Report

The manuscript describes synthesis of new 2-nitropyrroles functionalized at position N-1 as potential drugs against Trypanosoma cruzi, and their biological evaluation. This could be a nice contribution to medicinal chemistry, in the field of antiparasitic therapy, however, some improvements are needed.

The introduction is clear and comprehensive enough. In the section 2. Results and discussion - the results of the synthesis are clear and sufficiently discussed, but the Biological evaluation (part 2.2.) lacks discussion; very briefly only results are given. Therefore, this part should be expanded and it would be especially important to explain and discuss why (only) compound 18 was chosen for the intracellular Trypanosoma cruzi assay.

Line 142 – compound 3 not 13. It should also be corrected throughout the manuscript.

Line 158 – range not ranger.

Line 459 – it should be written what is exactly in the supplement.

Supplement – For all 1H and 13C NMR would be more useful if the region with peaks was shown expanded instead of showing spectra with full scales. At the moment, it is fine only for the spectrum on page 35.

1H NMR spectrum of compound 7 on page 6 – the first peak is not in the right descending order.

1H NMR Compound 13 on page 13 – the multiplet 4.13-4.08 is not assigned and integrated in the spectrum; the multiplet 1.21-1.25 is assigned and integrated (1H) in the spectrum, but nowhere is it described to which proton this belongs.

Author Response

Dear reviewer,

The presentation of the biological results was too succinct. The majority of the compounds were tested with the exception of compounds 20-22 (lack of solubility), but only compound 18 showed EC50 < 10 μM and selectivity against vero host cells. So, we initially focused our discussion on compound 18.

The biological part was expanded with complementary data:

  • cruzi EC50 for all compounds tested,
  • cruzi EC50 for Nifurtimox reference drug in Table 4
  • Vero cells EC50 for all compounds tested,
  • Dose-response curve data of compound 18 and nifurtimox in the manuscript,
  • Dose-response curve data of the other compound tested in the supplementary material,
  • Complete revision of the discussion: “These new derivatives were evaluated in vitro against the intracellular cruzi strain X10/7-Silvio by determining their 50% efficacy concentration (EC50) and were compared to the reference anti-T. cruzi drug nifurtimox. The in vitro 50% cytotoxic concentrations (CC50) were assessed on the CHO cell line and compared to a cytotoxic reference drug, doxorubicin. In vitro cytotoxicity against vero host cells was also assessed (vero cells EC50). The results are presented in Table 4. Regarding the effect on cell viability of the CHO cells, except for one compound (3), the other derivatives showed good cytotoxicity values ranging from 155.63 to 321.12 µM compared to the reference drug, doxorubicin (CC50 CHO = 4.65 µM). Three compounds showed no cytotoxicity against vero host cells at concentrations up to 50 µM (17-19). Of the 10 synthesized molecules, three showed poor DMSO solubility, limiting their in vitro anti-T. cruzi evaluation (20-22). Three compounds showed T. cruzi EC50 < 10 µM (4, 16 and 18). Only compound 18 exhibited activity against the parasite (EC50 = 3.6 ± 1.8 µM) without toxicity against host cells at concentrations up to 50 µM. However, the maximum effect of the latter compound (Figure 4) compared to the nifurtimox control (Figure 5) was relatively low (79% and 71% in two independent replicates), indicating that a substantial number of parasites survive 96h treatment. Whether longer duration treatments may eliminate all parasites remains to be explored. Structure-Activity Relationship (SAR) can be suggested from these biological results. The presence of a benzyl group seems to be unfavourable for anti-T. cruzi activity. Indeed, compounds 3 and 15 with respectively benzyl and meta-EWG 3-CF3 benzyl group exhibit no activity against T. cruzi (EC50 = 34.5 µM and 41.0 µM) whereas their analogue with phenyl (4) and meta-EWG 3-CF3 phenyl group (18) exhibit activity (EC50 < 10 µM). The meta-EWG 3-CF3 phenyl group gave the best anti-T. cruzi activity (18) while the meta-electro-donating OCH3 phenyl group (19) gave the lowest anti-T. cruzi activity (EC50 = 50.0 µM). An electron-withdrawing (EWG) group in the meta-position of the phenyl seems to be important for anti-T. cruzi activity.”

Line 142 –Compound 13 was corrected to compound 3 throughout the manuscript.

Line 158 – “Ranger” was corrected to “range”

Line 459 – Line 459 we added the content of the supporting information part: The supporting information are available online at www.mdpi.com/xxx/s1: 1H- for intermediates compounds 5-13; 1H and 13C-NMR for compounds 2-4 and 14-22; dose-response curve data for EC50 and CC50 calculations of compounds 3, 4, 15, 16, 17, 19.”

Supplement –For all all 1H and 13C NMR spectra the region with peaks was shown expanded as presented initially page 35.

For 1H NMR spectrum of compound 7 on page 6, the first peak was shifted to respect the right descending order

For 1H NMR compound 13, the multiplet 4.13-4.08 was assigned and integrated in the spectrum. The multiplet 1.21-1.25 was not assigned because it doesn’t correspond to a proton of the desired product. This impurity did not impact the next step and the spectrum of the final correspondant product 21 is clean.

Round 2

Reviewer 1 Report

This revised version of the manuscript substantially improved in quality and complemented in content can be accepted for publication in Molecules in present form.

Reviewer 3 Report

The authors have done good changes and added more info to the manuscript.

I don't feel that the manuscript is novel enough to warrant publication in a high-impact journals like Molecule.